

# Identification of SNPs associated with magnesium and sodium uptake and the effect of their accumulation on micro and macro nutrient levels in *Vitis vinifera*

Rachel P. Naegele[1], Jason P. Londo[2], Cheng Zou[3] and Peter Cousins[4]

[1] San Joaquin Valley Agricultural Sciences Center, USDA ARS, Parlier, CA, United States of America
[2] Grape Genetics Unit, USDA ARS, Geneva, NY, United States of America
[3] BRC Bioinformatics Facility, Institute of Biotechnology, Cornell University, Ithaca, NY, United States of America
[4] E&J Gallo Winery, Modesto, CA, United States of America

Corresponding author
Rachel P. Naegele,
rachel.naegele@usda.gov

## ABSTRACT

Macro and micro nutrient accumulation affects all stages of plant growth and development. When nutrient deficiencies or excesses occur, normal plant growth is altered resulting in symptoms such as leaf chlorosis, plant stunting or death. In grapes, few genomic regions associated with nutrient accumulation or deficiencies have been identified. Our study evaluated micro and macro nutrient concentrations in *Vitis vinifera* L. to identify associated SNPs using an association approach with genotype by sequencing data. Nutrient concentrations and foliar symptoms (leaf chlorosis and stunting) were compared among 249 $F_1$ *Vitis vinifera* individuals in 2015 and 2016. Foliar symptoms were consistent ($\geq$90%) between years and correlated with changes in nutrient concentrations of magnesium ($r = 0.65$ and $r = 0.38$ in 2015 and 2016, respectively), aluminum ($r = 0.24$ and $r = 0.49$), iron ($r = 0.21$ and $r = 0.49$), and sodium ($r = 0.32$ and $r = 0.21$). Single nucleotide polymorphisms associated with symptoms, sodium, and magnesium were detected on each chromosome with the exception of 5, 7 and 17 depending on the trait and genome used for analyses explaining up to 40% of the observed variation. Symptoms and magnesium concentration were primarily associated with SNPs on chromosome 3, while SNPs associated with increased sodium content were primarily found on chromosomes 11 and 18. Mean concentrations for each nutrient varied between years in the population between symptomatic and asymptomatic plants, but relative relationships were mostly consistent. These data suggest a complex relationship among foliar symptoms and micro and macro nutrients accumulating in grapevines.

## INTRODUCTION

Macro and micronutrients are essential for proper cell function and overall plant health. Macronutrients, those needed in large quantities by plants, include nitrogen, phosphorus, potassium, calcium, sulfur, and magnesium. These are largely present in the soil and

are readily available to plants depending on soil pH and moisture (*Maathuis, 2009*). Micronutrients, such as sodium, boron, iron, zinc, manganese and copper, are less prevalent in the soil, but small quantities are still necessary for plant growth and development. Nutrient levels fluctuate in the plant, and vary based on developmental stage, maturity, genotype, and tissue (*Benito et al., 2013*; *Pradubsuk & Davenport, 2010*).

Nutrient deficiencies often result from poor ion availability or uptake, leading to deformation of shoots or roots, uneven ripening of fruit, and chlorosis or necrosis of leaves. Leaf chlorosis is a common symptom of nutrient deficiency, as many macro and micronutrients contribute to chlorophyll production, enzyme and membrane stabilization and activation. Magnesium (Mg) is an important structural component of chlorophyll and a phosphorylizer or dephosphorilizer of compounds. Symptoms of Mg deficiency, such as interveinal chlorosis of the leaves, necrotic leaf spots, and root and shoot stunting can be induced by low levels of Mg or high levels of calcium (Ca), potassium (K) or other ions, which can alter Mg absorption (*Guo et al., 2016*; *Hermans & Verbruggen, 2005*; *Skinner & Matthews, 1990*; *Spiers & Braswell, 1994*).

Sodium (Na) can be used by plants in small quantities, but in excess, causes stunting, leaf tip burning, and leaf darkening (*Bernstein, 1975*). Leaf chlorosis, found in many nutrient deficiencies, is not a characteristic symptom of Na excess, except as a result of cation imbalances. These imbalances can be the result of substrate competition, as is the case with Mg, K and Ca, or can occur through changes in ion potential and turgor pressure (*Grattan & Grieve, 1992*; *Zhu, Liu & Xiong, 1998*). This complex relationship, while not well studied, varies among host species and type of salt ions (*Carbonell-Barrachina, Burlo-Carbonell & Mataix-Beneyto, 2008*; *Cordovilla et al., 1995*; *Volkmar, Hu & Steppuhn, 1998*). Complex relationships are also true among metal ions and nutrients in the soil. Aluminum (Al), a highly abundant metal in earth's crust, is one of the major factors limiting crop production in low pH soils (*Mossor-Pietraszewska, 2001*). Aluminum competes with other ions such as Mg or Ca for binding sites in the plant, leading to root deformation and nutrient deficiencies. It is often the lack of essential nutrients, and not the accumulation of toxic metals, that results in metal toxicity symptoms.

In grape, a perennial woody vine, nutrient fluctuations occur throughout the season with specific nutrient concentrations peaking during critical periods of development and growth. The composition and quantities of these nutrients can have drastic effects on fruit quality and plant health pre and postharvest (*Conradie, 1981*; *Conradie, 1992*; *Morris, Sims & Cawthon, 1983*; *Mpelasoka et al., 2003*; *Rogier et al., 2000*; *Schreiner, 2016*; *Williams, Maier & Bartlett, 2004*). In cultivated grape, *Vitis vinifera*, nutrient deficiencies are commonly observed in poor quality soils and can affect bud development, fruit yield, and quality (*Brancadoro et al., 1995*; *Tagliavini & Romboloa, 2001*). Fe and Mg are two of the most common deficiencies observed in grape, often observed as interveinal chlorosis (*Brancadoro et al., 1995*; *Conradie & Saayman, 1989*). Common nutrient excesses include Na and K (*Downton, 1977*; *Gong et al., 2015*; *Baneh, Hassani & Shaieste, 2014*) , though the severity of response can vary greatly depending on the genotype used and level of excess (*Kocsis & Walker, 2003*; *Poor et al., 2013*). However, foliar symptoms may also be the result of interactions among nutrients, and this has not been well-studied (*Shikhamany,*

*Chititrai & Chadha, 1988*; *Skinner & Matthews, 1990*). *Skinner & Matthews (1990)* found that adding phosphorous to the soil eliminated Mg deficiency symptoms and increased overall Mg concentrations.

Genotypic variation in nutrient levels is often caused by differences in the ability of a plant to uptake, accumulate, or metabolize nutrients (*Christensen, 1984*). Studies on the genetic control of nutrient accumulation in grape are limited, those that exist merely show the complexity surrounding nutrient absorption and their interactions (*Davies et al., 2006*; *Jimenez et al., 2007*; *Perez-Castro et al., 2012*; *Primikiros & Roubelakis-Angelakis, 2001*). QTL analyses have identified regions associated with Fe and Na tolerance and Mg deficiency. For Fe tolerance, a major QTL located on chromosome 13 explained up to 50% of the phenotypic variation in root and shoot biomass over two years using a *Vitis* inter-specific cross between Cabernet Sauvignon (*V. vinifera*) and Gloire de Montpellier (*V. riparia*) under chlorosing conditions. Minor effect QTL were also detected on chromosomes 5, 9, 18, 19 with variation evident between years (*Bert et al., 2013*). Two QTL on chromosomes 11 and 13 were associated with Fe concentration in grafted plants only. An interspecific-hybrid population between two rootstocks was evaluated for leaf sodium exclusion. Na leaf concentrations were found to be associated with a block of 538 genes located on chromosome 11 explaining 72% of the variation (*Henderson et al., 2018*). The authors characterized the proteins from four different alleles of high-affinity potassium transporters, and found allelic variants affected Na accumulation. For Mg deficiency, leaf symptoms and Mg concentrations were negatively correlated ($r = -0.52$), and it was determined that deficiency was controlled by a major QTL accounting for approximately 55% of the variation located on linkage group 11 (*Mandl et al., 2006*). Based on unstable inheritance in later generations, it was postulated that highly symptomatic plants were the result of an interaction between alleles from both progenitors. However, this study did not evaluate the levels of other elements such as P, K, and Ca which are known to affect Mg absorption and allocation. Each of these studies identified QTL using inter-specific crosses.

In grape, few studies have examined the genetics of nutrient absorption and concentrations and its relationship to phenotypic variation despite importance in plant development and fruit quality. Mapping families remain a useful tool for understanding the genetic architecture of complex traits, such as nutrient balance, and we observed symptoms initially believed to be Mg over-accumulation in an $F_1$ breeding population derived from a cross between two *V. vinifera* cultivars, 'Verdejo' and "Gewürztraminer". Leveraging the structure of this $F_1$ population, the objectives of this study were to determine the relationship between nutrients and visible symptoms, heritability and segregation, identify genomic regions associated with magnesium, sodium, and other macro and micro nutrients accumulation in *Vitis vinifera* L., and compare SNP detection across two reference genomes.

## MATERIALS AND METHODS

### Material and nutrient analyses

Two hundred forty-nine seedlings of a segregating *Vitis vinifera* $F_1$ breeding population derived from two heterozygous grape varieties, 'Verdejo' × 'Gewürztraminer' (V×T)

were transplanted in June 2013 into a research plot in Ripperdan, CA (soil type = Cajon loamy sand, Dinuba-El Peco fine sandy loam, Pachappa sandy loam, slightly—moderately saline-alkali; pH = 7.9). All vines were own rooted with no grafting. Row spacing was set at 1.22 m with 2.44 m between rows. Seedlings were trained and managed according to standard grower practices. Plants were fertilized with N, P, and K at rates of 14.5, 18.4, and 12.9 kg/hectare, respectively in 2015 and 18.1, 23.1, and 16.1 kg/hectare in 2016. Fertilization was performed according to industry standard practices; fertigation by applying a liquid fertilizer solution through the drip irrigation every two weeks from the time of fruit set. Lateral shoots were removed from the trunk during establishment and vines were trained to a unilateral cordon and spur-pruned. Plants were visually assessed for foliar symptoms in August (2015) and September (2016) using a 1 (present) or 0 (absence) rating where symptoms were plant stunting and/or leaf chlorosis (Fig. 1). Plant stunting and leaf chlorosis were evaluated separately. For nutrient analysis fully expanded whole leaf (petiole and blade) samples were collected from each vine. Due to variability between genotypes, equivalent leaf volume was collected, typically between 15–25 mature leaves. The leaves were sampled from fertile (fruiting) shoots into brown paper bags and air dried indoors at 22 °C. Once dry, the leaves were submitted to A & L Western Laboratories (Modesto, CA) for nutrient analyses in October 2015 and in October 2016. Nutrient concentrations were measured for nitrogen (N), sulfur (S), phosphorus (P), potassium (K), magnesium (Mg), calcium (Ca), sodium (Na), iron (Fe), aluminum (Al), manganese (Mn), boron (B), copper (Cu), and zinc (Zn). N was measured using automated combustion at 900 °C. S, P, K, Mn, Ca, Na, Fe, Al, Mg, B, Cu and Zn were measured using nitric/hydrochloric acid digestion using a microwave, analysis was by inductively coupled plasma spectrometry (ICP) as detailed by The North American Proficiency Testing Program (*Black, 1965*; naptprogram.org). N, S, P, K, Mg, Ca, and Na were reported as a percent of dry matter (% dm). Fe, Al, Mn, B, Cu and Zn were reported as parts per million (ppm). At the end of the study, a subset of symptomatic and asymptomatic vines was removed and evaluated for root stunting.

## Statistical analysis

Nutrient data were analyzed using JMP v12 statistical software (SAS Institute, Cary, NC) for normality (Shapiro-Wilk W Test), analysis of variance (ANOVA), hierarchical clustering, and correlations for relationships within and between years. Plant symptoms were analyzed as marginal chlorosis only, stunting only, or combined (stunting and/or chlorosis). Data for Zn, Na, P, and Mn were log transformed, S, Mg, Fe, B, and Al were log10 transformed, and Ca and K were square root transformed to fulfill assumptions of normality. Significant differences in nutrient concentrations between years or genotypes were determined using Tukey's Honest Significant Difference (HSD) ($P \geq 0.05$). Correlations were determined using Pearson's correlation coefficient (r) on the transformed data. Hierarchical clustering was determined using the Ward method on standardized data. Broad sense heritability (H) was calculated based on mean square values using the one location across two years formula modified from *Fehr (1987)* by *Wang, Karle & Iezzoni (2000)* with confidence intervals estimated by *Knapp, Stroup & Ross (1985)*. Best Linear Unbiased Predictors (BLUPs) were calculated with the lme4 package in R (v4.0.2 *R Core Development Team,*

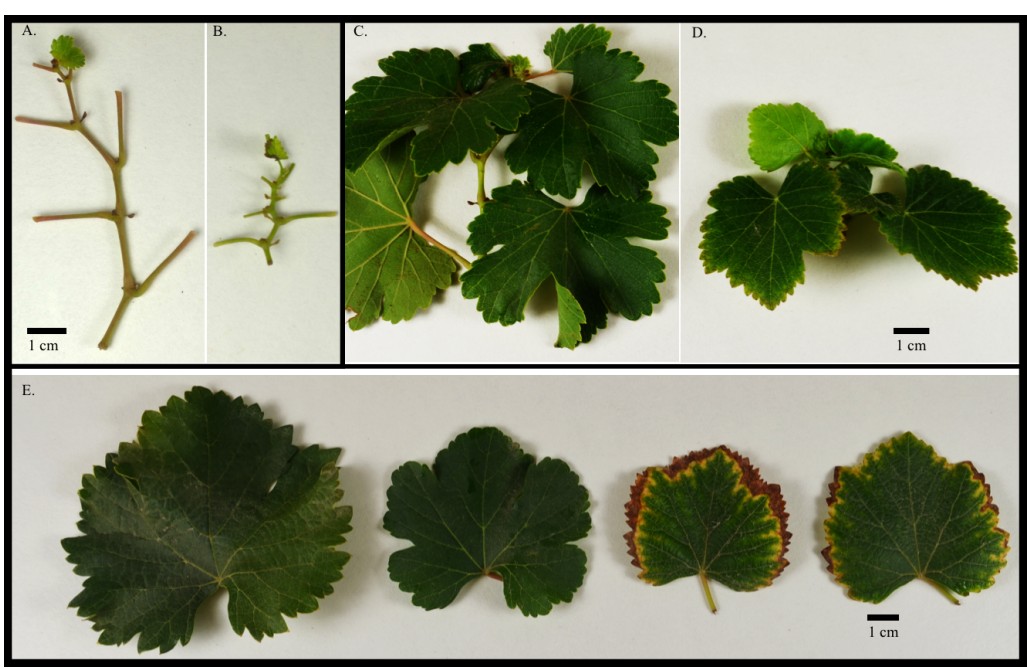

**Figure 1** **(A) Asymptomatic and (B) symptomatic (stunted) grape stem internodes and (C–D) leaves from an F1 V. vinifera population; (E) Asymptomatic and symptomatic (marginal leaf chlorosis and stunted) leaves.**

2017) using nutrient concentration, genotype, and year as random effects (*Henderson, 1975*; *Liu, Rong & Liu, 2008*; *Merk, 2011*). Principal component analysis for the population was calculated using the BLUPs for each nutrient concentration within JMP12.0.1.

## Genotyping by sequencing and mapping of significant SNP associations

Young grape leaves were collected from each $F_1$ progeny grapevine in July 2015 and genomic DNA extracted using the Qiagen genomic DNA extraction kit (Qiagen, Inc Valencia, CA). Genomic DNA was sent to the UC Davis Genome Center's DNA Technologies and Expression Analysis Cores (University of California, Davis) for quality analysis, restriction enzyme digestion (ApeK1), library preparation and Illumina Hi-seq 3000 sequencing. Sequencing coverage was approximately 2.7 million reads per sample. Genotyping by sequencing (GBS) data was analyzed using the Tassel 5.0 GBSv2 pipeline (*Bradbury et al., 2007*). The table grape/raisin genome of Thompson Seedless was used, in addition to the wine grape-derived inbred genome of PN40024, to capture some of the variability in SNP detection between reference genomes. Quality ($\leq 20$) and length ($\geq 20$ bp) filtered and trimmed reads were aligned to the Thompson Seedless genome (*Genova et al., 2014*; *Patel et al., 2018*) and the PN40024 12xv2 genome (*Canaguier et al., 2017*) using BWA (*Li & Durbin, 2010*). Identified SNPs were further filtered for frequency of minor (0.20) and major ($\geq 0.35$) allele frequencies, missing data ($\leq 10\%$), and sequencing depth ($\geq 5$ reads (Thompson Seedless) or $\geq 10$ reads (PN40024)) using vcftools 0.1.15

(*Danecek et al., 2011*). For the PN40024 genome, an additional filtering step to thin SNPs based on physical position to a minimum of 50 bp between sites was completed. A panel of 10,122 and 3,997 filtered SNPs (Thompson Seedless and PN40024, respectively) were used for genome-wide association analyses for each of the ions measured (Table 1). A kinship matrix was estimated in Tassel v 5.2.43 and used in a MLM (mixed linear model) implemented in the software GAPIT v2 for nutrient trait analyses within R statistical analysis software (*Lipka et al., 2012*; *Tang et al., 2015*; *R Core Development Team, 2017*). For binary traits (stunting, chlorosis, and combined symptoms), 2 principal components with P3D were used for analyses implemented within Tassel. Significance of a SNP was based on a $P$ value $\leq 0.05$ and a false discovery rate (FDR) $\leq 0.05$. Linkage decay was estimated using PopLDdecay for the PN40024 genome (*Zhang et al., 2019*). In brief, pairwise $r^2$ for SNPs within 5 Mb were calculated using PopLDdecay, then the median $r^2$ for every 10 kb window were calculated (*Zhang et al., 2019*). The continuity of the Thompson Seedless genome is of lower quality than that of PN40024. Thus to cross reference significant SNPs detected in each reference genome, PN40024 was used as a coordinate reference. GBS tags with significant SNPs detected in Thompson seedless and their associated flanking sequence were mapped back to PN40024 to unify coordinates. Sequences were mapped using default parameters for short read alignment using Minimap2 (*Li, 2018*). Uniquely mapped primary alignments with quality higher than 40 were kept in the genome coordinate liftover. Manhattan plots for chromosome 3 were produced using the CMplot package in R (https://github.com/YinLiLin/R-CMplot). This method allowed for the ordering of SNPs in the same coordinate reference and comparison between distributions.

Functional annotation of genes associated with SNPs was determined using Blast2GO v 5.2.5 based on the nonredundant database from NCBI, and protein databases from Uniprot and Swissprot (*Gotz et al., 2008*; accessed June 2018).

## RESULTS

### Field symptoms

There were 60 and 52 individual vines exhibiting symptoms, while 189 and 197 did not exhibit symptoms in 2015 and 2016, respectively, roughly following a 3:1 segregation. The parents, not grown at the time of this study, had not previously displayed any symptoms of nutrient imbalances at this location under similar fertilization regimes. Symptoms observed in the $F_1$ progeny included leaf, internode, and petiole stunting, as well as marginal leaf chlorosis and necrosis (Fig. 1). A subset of symptomatic and asymptomatic plants evaluated for root stunting showed no visible differences (*data not shown*). Presence of symptoms (stunting or chlorosis) was consistent between 2015 and 2016 for most vines (>90%). Only eighteen vines had symptoms in 2015, but were asymptomatic in 2016. Another two genotypes had no symptoms in 2015, but were symptomatic in 2016. When each symptom was evaluated individually, stunting and marginal chlorosis symptoms were consistent among plants in both years (>80%).

**Table 1  Single Nucleotide Polymorphism distribution across the Thompson Seedless and PN40024 reference genomes.**

| Chromosome | Number of SNPs | |
|---|---|---|
| | Thompson Seedless | PN40024 |
| 1 | 608 | 189 |
| 2 | 271 | 107 |
| 3 | 385 | 180 |
| 4 | 543 | 225 |
| 5 | 746 | 322 |
| 6 | 479 | 207 |
| 7 | 663 | 240 |
| 8 | 640 | 260 |
| 9 | 476 | 214 |
| 10 | 676 | 253 |
| 11 | 279 | 113 |
| 12 | 624 | 257 |
| 13 | 609 | 241 |
| 14 | 753 | 278 |
| 15 | 436 | 137 |
| 16 | 418 | 180 |
| 17 | 463 | 172 |
| 18 | 655 | 283 |
| 19 | 398 | 139 |
| Total | 10,122 | 3,997 |

## Nutrient compilation

For the population, significant differences were detected in nutrient concentrations between 2015 and 2016 (Table 2). Large increases in Al, Fe, Mg, Zn, Mn and Ca concentrations were observed in leaf tissue between samples collected in 2015 and 2016. Mean values for nutrient concentrations varied for most of the ions evaluated among individuals in the V×T population (Fig. S1; Table S1). A decrease in N, P, K, and B leaf nutrient concentrations was observed from 2015 to 2016. When symptomatic and asymptomatic plants were analyzed separately, differences in nutrient concentration were detected in 2015 and 2016 (Table 3). Higher levels of Mg, Na, Al, and Fe were observed in symptomatic plants in both years, while a decrease in N was observed. Na, P, Cu, Mn, N, S, and Ca concentrations in 2015 or 2016 had no calculable heritability. For Mg ($H = 0.34$; confidence intervals (CI): 0.18–0.46), B ($H = 0.44$; CI [0.32–0.55]), K ($H = 0.21$; CI [0.02–0.36]), Al ($H = 0.12$; CI [0–0].28), and Fe ($H = 0.27$; CI [0.10–0.41]) broad sense heritability was moderate to low.

## Nutrient concentration and symptom correlations

Symptoms (marginal chlorosis, stunting or both) were positively correlated with Na, Mg, Fe and Al concentrations, and negatively correlated with N across both years (Fig. 2, Table 4, Fig. S2). In 2015 Mg concentrations ($r = 0.6146$) and in 2016 Al concentrations

**Table 2  Mean nutrient concentrations from combined leaf and petiole samples collected in 2015 and 2016 from an F₁ *Vitis* population.**

| Nutrient | Unit[a] | Population Mean ± (StD) | | Normal range[b] |
|---|---|---|---|---|
| | | 2015 | 2016 | (petioles) |
| N | % dm | 2.19 ± 0.25[*] | 1.62 ± 0.26 | 0.8–1.2 |
| S | % dm | 0.18 ± 0.02[*] | 0.17 ± 0.03 | – |
| P | % dm | 0.17 ± 0.03[*] | 0.13 ± 0.03 | 0.14–0.30 |
| K | % dm | 0.94 ± 0.24[*] | 0.70 ± 0.24 | 1.2–2.0 |
| Mg | % dm | 0.69 ± 0.17[*] | 0.83 ± 0.15 | 0.35–0.75 |
| Ca | % dm | 2.44 ± 0.44[*] | 3.27 ± 0.56 | 1–2 |
| Na | % dm | 0.02 ± 0.02[*] | 0.05 ± 0.03 | – |
| Fe | ppm | 400.41 ± 86.27[*] | 484.20 ± 175.14 | 30–100 |
| Al | ppm | 238.05 ± 57.14[*] | 315.67 ± 112.02 | – |
| Mn | ppm | 59.31 ± 13.08[*] | 69.53 ± 18.69 | 100–1000 |
| B | ppm | 50.81 ± 14.60[*] | 42.00 ± 16.02 | 25–50 |
| Cu | ppm | 6.20 ± 2.43 | 6.12 ± 1.25 | 5–15 |
| Zn | ppm | 20.78 ± 3.95[*] | 26.12 ± 5.88 | 30–60 |

Notes.
[a]Units of measurement for each micro or macronutrient analyzed as percent dry matter (% dm) or parts per million (ppm).
[b]Typical range for petiole concentrations for *Vitis* cultivars selected from *Bates & Wolf (2008)*.
*indicates a significant difference ($P \leq 0.05$) in concentration between 2015 and 2016.

**Table 3  Population means for grapevine nutrient concentrations.**

| Ion | Unit[a] | 2015 | | 2016 | |
|---|---|---|---|---|---|
| | | No Sym[b] | Symp | No Symp | Symp |
| N | % dm | 2.22 ± 0.23[*] | 2.09 ± 0.27 | 1.63 ± 0.26[*] | 1.53 ± 0.22 |
| S | % dm | 0.18 ± 0.02[*] | 0.17 ± 0.02 | 0.17 ± 0.03 | 0.17 ± 0.02 |
| P | % dm | 0.17 ± 0.03 | 0.17 ± 0.02 | 0.13 ± 0.03[*] | 0.15 ± 0.03 |
| K | % dm | 0.93 ± 0.24 | 0.96 ± 0.25 | 0.66 ± 0.23[*] | 0.86 ± 0.26 |
| Mg | % dm | 0.63 ± 0.11[*] | 0.89 ± 0.19 | 0.80 ± 0.14[*] | 0.95 ± 0.14 |
| Ca | % dm | 2.44 ± 0.43 | 2.45 ± 0.44 | 3.37 ± 0.54[*] | 2.86 ± 0.45 |
| Na | % dm | 0.02 ± 0.02[*] | 0.03 ± 0.02 | 0.04 ± 0.02[*] | 0.06 ± 0.05 |
| Fe | ppm | 389.85 ± 84.44[*] | 431.57 ± 85.68 | 443.28 ± 139.08[*] | 660.06 ± 205.68 |
| Al | ppm | 230.19 ± 55.58[*] | 261.25 ± 55.76 | 289.02 ± 89.17[*] | 430.21 ± 128.03 |
| Mn | ppm | 58.61 ± 12.69 | 61.35 ± 14.15 | 70.70 ± 19.17[*] | 64.49 ± 15.63 |
| B | ppm | 50.40 ± 14.39 | 52.02 ± 15.24 | 41.46 ± 16.61 | 44.34 ± 13.07 |
| Cu | ppm | 6.17 ± 2.60 | 6.32 ± 1.87 | 5.95 ± 1.20[*] | 6.85 ± 1.20 |
| Zn | ppm | 20.96 ± 3.90 | 20.25 ± 4.08 | 25.77 ± 5.20 | 27.62 ± 8.11 |

Notes.
[a]Units of measurement for each micro or macronutrient analyzed as percent dry matter (% dm) or parts per million (ppm).
[b]Non symptomatic (No sym) and Symptomatic (Symp) plants.
*indicates a significant difference between symptomatic and asymptomatic plants.

($r = 0.4805$) had the highest correlation with observed vineyard symptoms (Table 4). Na and Fe concentrations were also correlated with symptoms in both years, though at lower r values. In 2015, a significant negative correlation between S concentration (22%) and symptoms was observed, but not in 2016 (Table S2). In 2016, there was a significant

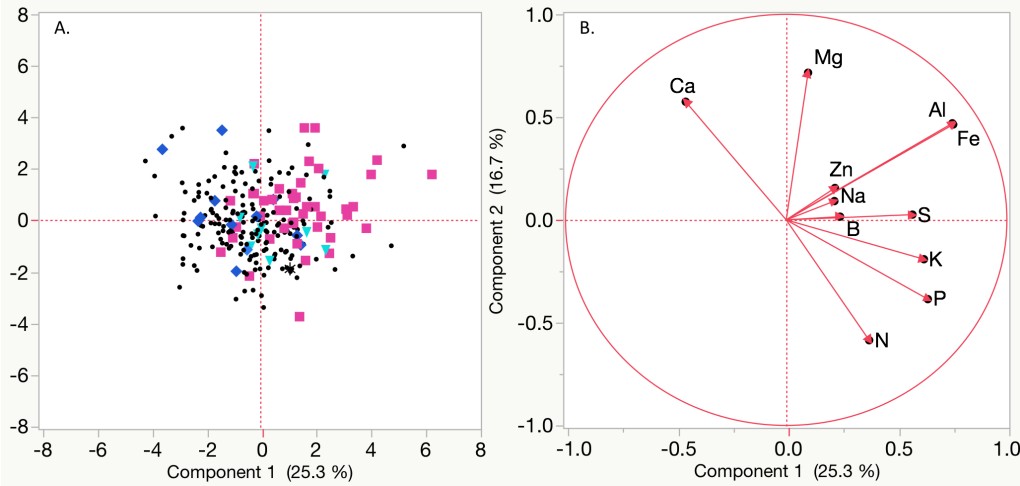

**Figure 2** **Principal Component Analysis (PCA) based on mean Best Linear Unbiased Predictors (BLUPs) for nutrient concentrations of magnesium (Mg), sodium (Na), aluminum (Al), iron (Fe), potassium (K), phosphorous (P), nitrogen (N), sulfur (S), zinc (Zn), boron (B), and calcium (Ca).** (A) Black circles represent non symptomatic plants, blue diamonds indicate plants that exhibited symptoms (only marginal leaf chlorosis (MC) or stunting (SL)) in only one year, teal triangles indicate plants that had symptoms (MC and SL) in only one year, black asterisks represent plants that had SL or MC for both years, and pink squares indicate vines with stunting and leaf chlorosis in both years. (B) Vectors for each nutrient based on BLUPs.

negative correlation between P, Cu, and K content and symptoms (Table S2). Other significant correlations among nutrients included positive correlations ($r \geq 0.30$) between N, P, and S in 2015 and S, P, and K in 2015 and 2016 (Table S2). A strong positive correlation was also detected for both Mn and Ca with Mg in 2015, but not 2016. Correlation with symptoms were observed for other nutrients, but were not consistent between years. When comparing nutrient concentrations from 2015 to 2016, most nutrients had low to moderate ($r = 0.2 - 0.4$) correlation, with the exception of copper ($r = -0.0199$) (Table S3). Nutrient ratios were examined between years for potential significant correlations with symptoms. Most ratios did not show consistent differences in values between years for symptomatic and non symptomatic vines (Table S4).

## Marker-trait associations

Linkage disequilibrium half-decay distance was estimated to be 100 Kbp (Fig. S3). Genome-wide associations identified several chromosomes associated with differences in the ions evaluated (Fig. S4, Fig. 3). Significant positive associations between SNPs on chromosome 3 and Mg concentration were detected in 2015 for both the Thompson Seedless and PN40024 genomes. SNPs associated with Mg levels explained approximately 6% of trait variation (Table S5). In 2015, 6 SNPs were detected when aligned to the PN40024 genome while 4 SNPs were detected when aligned to Thompson Seedless. No SNPs associated with Mg accumulation were identified in 2016 with either genome at the $P = 0.05$ FDR level. Only 1 genic SNPs associated with Mg concentration was identified and was co-associated with SNPs identified for marginal leaf chlorosis and stunting (Tables S6 and S7). In the

**Table 4 Correlation (r) among nutrient concentrations in 2015 (gray) and 2016 (white) from grape vines.**

|  | Mg | Ca | Na | Fe | Al | N | Sym[a] |
|---|---|---|---|---|---|---|---|
| Mg | – | 0.4414*** | 0.3175*** | 0.1663** | 0.2220** | −0.3552*** | 0.6146*** |
| Ca | 0.4123*** | – | 0.1414* | NS | NS | −0.3394*** | NS |
| Na | 0.2134** | NS | – | 0.1847* | 0.1601* | NS | 0.2973*** |
| Fe | NS | −0.2168** | 0.1850* | – | 0.9675*** | −0.1512* | 0.2149** |
| Al | 0.1385* | −0.1958* | 0.1919* | 0.9841*** | – | −0.2290** | 0.2406** |
| N | −0.2852* | −0.2822*** | NS | NS | NS | – | −0.2275** |
| Sym | 0.3200*** | −0.3619*** | 0.2436*** | 0.4724*** | 0.4805*** | −0.1679* | – |

**Notes.**

[a] Symptoms

*$P \leq 0.05$

**$P < 0.001$

***$P < 0.0001$

NS, not significant.

Thompson Seedless genome, a small block of SNPs (S3_21825918 , S3_21825925, and S3_21825966), spanning 48 bp, located in a ∼7,000 bp intra genomic region, explained ∼18% of the variation associated with symptoms in 2015. A BLAST search of the region did not identify any significant alignments with any genes (predicted, putative or known) in *Vitis* or other species.

SNPs associated with Na concentration were detected on chromosomes 11, 12, 13, and 18 had both negative and positive allelic effects ranging from 8 to 13% of the observed variation in the Thompson Seedless genome. Three of the SNPs (S11_16152632, S18_25745134, and S18_25745143) were detected in 2015 and 2016. None of the SNPs associated with Na concentration in 2015 were located in predicted or known genes. Only a single SNP identified in 2016 (S11_11635675) was found in an annotated gene (Table S6). In the PN40024 genome, 11 SNPs with positive allelic effects were detected across chromosomes 3, 11, 15, and 17 in 2016. Individual SNPs explained 6 to 11% of the variation observed. Two of the SNPs on chromosomes 3 and 11 were also associated with Na concentrations in 2015, but were not significant when adjusted for an FDR of 0.05. Using the PN40024 genome annotation, 8 genes were associated with Na accumulation. These included genes putatively involved metabolism and transport. Four of the genes had no functional annotation ascribed (Table S5). One gene, Vitvi11g01139, was associated with Na accumulation and annotated as a Clathrin assembly protein, which is a class of proteins involved in macromolecule transportation. No significant SNPs were identified for any of the remaining ions measured.

For marginal chlorosis, a total of 151 SNPs, 57 in 2015 and 94 in 2016 were detected across 2015 and 2016, respectively in the Thompson Seedless genome. In 2015, 33 of the identified SNPs were located in genes, while in 2016, 20 SNPs were located in genes (Table S5). Sixteen of the genic SNPs were shared between years. Individual SNP effects on the Thompson Seedless genome were both negative and positive and ranged from 16 to 33%. Many of the SNPs identified in only a single year were located in genes with multiple SNPs associated with the trait (Table S6). When aligned to the PN40024 genome, a total of 29 and 55 SNPs in 2015 and 2016, respectively associated with marginal chlorosis were
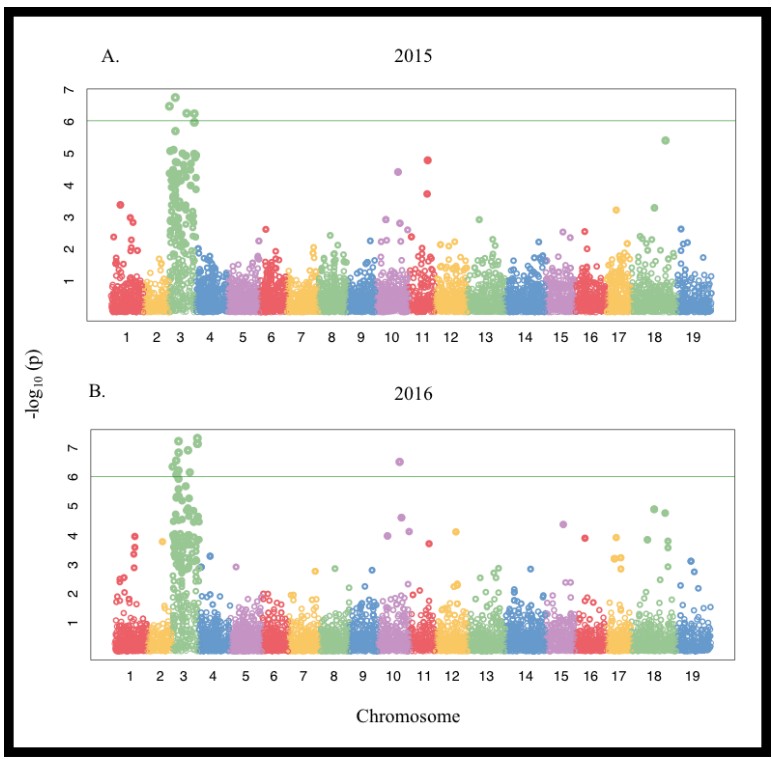

**Figure 3** Manhattan plot of Single Nucleotide Polymorphisms (SNPs) associated with marginal leaf chlorosis/burn and stunting symptoms in a *V. vinifera* $F_1$ segregating population in (A) 2015 and (B) 2016 aligned to the PN40024 genome. The green horizontal line denotes the genome-wide significance threshold at $P < 1.0 \times 10^{-6}$.

detected. Marginal chlorosis was primarily associated with SNPs on chromosome 3 with other SNPs detected on chromosomes 6, 9, 13, 16, 17 and 19. In 2015, 23 of the identified SNPs were located in genes, while in 2016, 22 SNPs were located in genes, 22 SNPs were shared between years. Similar to the SNPs detected in Thompson Seedless genome, many SNPs were consistent across both years on chromosomes 3 and 16.

A total of 37 and 71 SNPs associated with stunting were identified in 2015 and 2016, respectively when mapped to the Thompson Seedless genome. SNP effects varied from 3 up to 40%. Highest effect SNPs were detected on chromosome 3 with smaller effect SNPs located on chromosomes 18 (2015) and 1, 2, 10, 11, 12, 16 and 18 (2016). Twenty-four and 34 SNPs were associated with genes in 2015 and 2016, respectively. Nineteen of the genic SNPs associated with plant stunting were shared between years. The majority of single-year SNPs were identified in 2016 (Table S5). When mapped to the PN40024 genome, a total of 34 and 44 SNPs were detected in 2015 and 2016, respectively. Only 34 and 35 SNPs associated with stunting were located in genes for 2015 and 2016, respectively of which 29 were shared between 2015 and 2016.

No ion transport pathways were associated with symptom-associated SNPs based on the Thompson Seedless annotation, however approximately 50% of the genes had putative catalytic activity and 50% had binding activity (Fig. S5). None of the genes identified across

**Table 5  Genic Single Nucleotide Polymorphisms (SNPs) associated with symptoms (marginal leaf chlorosis and stunting) using the PN40024 genome annotation and NCBI in 2015 and 2016 in an F₁ population of *V. vinifera*.**

| Chr[a] | Gene[b] | SNP | Effect[c] | Putative function[d] |
|---|---|---|---|---|
| 3 | Vitvi03g00380 | S3_4196400 | 23–25% | Unknown |
| | Vitvi03g01518 | S3_4201002 | (-)25% | PREDICTED: uncharacterized protein |
| | Vitvi03g00384 | S3_4208958 | 21–25% | Integral membrane protein |
| | Vitvi03g00384 | S3_4209015 | 21–25% | Integral membrane protein |
| | Vitvi03g00430 | S3_4637832 | (-)23–27% | Dof zinc finger protein DOF5.8 |
| | Vitvi03g00520 | S3_5653914 | 24–27% | Basic helix-loop-helix (bHLH) family |
| | Vitvi03g00534 | S3_5852953 | 23–24% | ABA-specific glucosyltransferase |
| | Vitvi03g00543 | S3_5986778 | (-)26–30% | DNA-directed RNA polymerase II |
| | Vitvi03g00560 | S3_6167883 | 29–31% | UNC-50 |
| | Vitvi03g00583 | S3_6554413 | (-)23% | TIP41 |
| | Vitvi03g00603 | S3_6823070 | 29–32% | R protein MLA10 |
| | Vitvi03g00688 | S3_7815436 | 33% | Hypothetical protein |
| | Vitvi03g00688 | S3_7815488 | (-)33% | |
| | Vitvi03g00777 | S3_9374358 | 25–29% | EMB2758 (embryo defective 2758) |
| | Vitvi03g01012 | S3_14786293 | (-)29% | No hit |
| | Vitvi03g01792 | S3_16473090 | (-)26–31% | Peru 1 |

**Notes.**
[a] Chromosome
[b] Putative grape gene based on the PN40024 v2 genome (*Canaguier et al., 2017*).
[c] Percent of the variation explained by a SNP.
[d] Functional annotation based on PN40024 genome v3 annotation (*Canaguier et al., 2017*).

both years and associated with symptoms were putative transporters, but were instead involved in processes such as oxidation, transcription, development, and stress response (Table 6). When symptom-associated SNPs located in genes based on the PN40024 annotation were evaluated for putative activity, stress response, transcription, growth and development, and metabolic pathways were all represented similar to the Thompson Seedless genome. In addition, SNPs were also detected in several genes related to sugar and nutrient transport. One SNP associated with leaf stunting in both years and symptoms was associated with Calcium ion binding (Vivi03g00243).

The majority of symptom-associated SNPs were found on chromosome 3 in both genomes, but each genome has a unique coordinate system. Therefore, we performed a consolidated genome analysis by mapping significant SNPs in Thompson seedless and their flanking regions to the PN40024 genome in order to order SNP coordinates and look for overlap between references (Fig. 4). When chromosome 3 assemblies were consolidated, a shared cluster of SNPs with significant association with symptoms was observed ~7.5 Mb and a lesser cluster around ~15 Mb. When aligned to the Thompson Seedless genome, symptom-associated SNPs located within genes shared across years were predominantly found on chromosome 3 with an additional SNP located on chromosome 10 (Table 6; Table S6). Significant single year SNPs, including those associated in genes, were identified on chromosomes 1, 2, 3, 4, 10, 11, 12, 16, and 18 (Table S5). When symptoms were combined, 28 SNPs were shared across both years, and 29 were only identified in a single

**Table 6** Genic Single Nucleotide Polymorphisms (SNPs) associated with symptoms using the Thompson Seedless genome annotation in 2015 and 2016 in an $F_1$ population of *V. vinifera*.

| Chr[a] | Gene[b] | SNP | Effect[c] | Putative function[d] |
|---|---|---|---|---|
| 10 | g1087 | S10_19692199 | 6-9% | Polyphenol oxidase |
| 3 | g1405 | S3_149721 | 9% | Probable beta-D-xylosidase 5 |
| | g1407 | S3_265316 | 5-6% | Uncharacterized protein LOC109124260 |
| | g1462 | S3_921272 | 6-7% | FAD-linked sulfhydryl oxidase ERV1 |
| | g1519 | S3_2023880 | 5% | HTH-type transcript regulator protein ptxE |
| | g1523 | S3_2032939 | 4-5% | CASP-like protein 5C1 |
| | | S3_2032882 | 4-6% | |
| | g1599 | S3_2800042 | 6% | Glycoside hydrolase, family 10 |
| | g1629 | S3_3258173 | 6-9% | At4g33990 |
| | g1632 | S3_3366650 | 7-9% | Oxysterol-binding protein 5 |
| | | S3_3366649 | 5-7% | |
| | g1658 | S3_3935538 | 7-9% | Polyphenol oxidase |
| | g1676 | S3_4305461 | 5% | Myb-binding protein 1A |
| | g1689 | S3_4773185 | 5% | Scopoletin glucosyltransferase-like |
| | g1736 | S3_5569343 | 7% | Receptor-like protein kinase HAIKU2 |
| | g1784 | S3_6826327 | 5-7% | Os01g0234100-like isoform X1 |
| | g1882 | S3_8415245 | 4-6% | Classical arabinogalactan protein 9 |
| | g1999 | S3_10174220 | 4-6% | Dof zinc finger protein DOF3.4-like |
| | g2046 | S3_11599143 | 7-8% | CSC1-like protein HYP1 isoform X1 |
| | g2086 | S3_12359712 | 7% | Protein unc-50 homolog |
| | g2129 | S3_13316810 | 5-6% | Serine/threonine-protein kinase BLUS1 like |
| | g2137 | S3_13577682 | 6-7% | Exocyst complex component SEC6 |
| | g2185 | S3_14537967 | 7% | Hypothetical protein VITISV_042288 |
| | g2192 | S3_14638969 | 5-6% | E3 ubiquitin-protein ligase MBR2 iso X1 |
| | g2217 | S3_15151065 | 9-10% | Mitochondrial Rho GTPase 1-like |
| | g2363 | S3_18838966 | 5-7% | DNA-directed RNA poly II, IV, V sub 3 |
| | g2581 | S3_22457867 | 6-7% | D-3-phosphoglycerate dehydrogenase 1 like |
| | g2622 | S3_23241926 | 6% | KH domain-containing protein HEN4 |

**Notes.**
[a] Chromosome
[b] Putative grape gene based on the Thompson Seedless genome (*Patel et al., 2018*).
[c] Percent of the variation explained by a SNP.
[d] Putative function based on BLAST2GO annotation (*Gotz et al., 2008*).

year (Table S5). When aligned to the PN40024 genome, significant SNPs were detected on chromosomes 3, 6, 11, 16, and 19. Most of the identified SNPs were located on chromosome 3, and 21 were shared between 2015 and 2016 (Table 5). Twenty SNPs were only detected in a single year, with the majority identified in 2016 (15). Twenty-nine SNPs found on chromosome 3 were shared between years 1 and 2. Sixteen genic SNPs associated with the combined symptoms were identified in both 2015 and 2016 (Table 5).
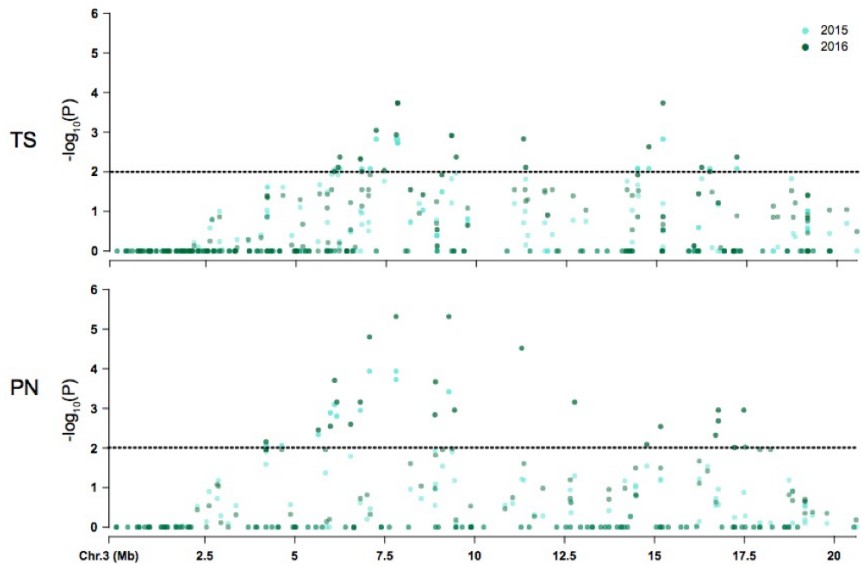

**Figure 4** **Coordinate comparison of the distribution of non significant and significant SNPs identified on chromosome 3 between Thompson Seedless (TS) and PN40024 (PN) reference genomes.** The dotted line denotes significance at $P = 0.05$.

# DISCUSSION

Proper macro and micro nutrient accumulation in grapevines is a perennial concern for growers, particularly in regions with marginal soils. Deficiencies, overaccumulations, or mis-partitioning of nutrients can result in economic losses in yield and fruit quality, and occasionally cause plant death. In our study, a heterozygous segregating *Vitis vinifera* F$_1$ population ('Verdejo' x 'Gewürztraminer'; denoted as V×T) segregating for foliar symptoms was evaluated for micro and macro nutrient and ion (N, S, P, K, Mg, Ca, Na, Fe, Al, Mn, B, Cu, and Zn) concentrations and symptom-associated SNPs. We chose to utilize a GWAS style approach to detect significantly associated SNPs amongst the progeny, similar to work by *Zou et al. (2020)* demonstrating genome-wide marker association with flower sex (2020) rather than pursue a traditional QTL approach. This approach allowed us to evaluate the genetic architecture of leaf symptoms through the high SNP detection produced by using next-generation GBS methods. In the case of Na, this approach enabled the detection of significant SNPs, even when low or no heritability was calculated, likely due to the low Na concentrations observed (0.0 to 0.2 % dm).

Normal nutrient ranges for plants vary depending on environment, variety, maturity, tissue, plant age, and developmental stage making comparisons among studies difficult even when using the same cultivar (*Benito et al., 2013*; *Pradubsuk & Davenport, 2010*; *Conradie, 1992*; *Schreiner, Scagel & Baham, 2006*; *Schreiner, 2016*). This difficulty is exemplified by the results presented here, where significant changes in ion concentrations varied in the two years of the study. Most ions showed an increase in concentration from 2015 to 2016, with the exception of B, P, N, K despite higher levels being applied in 2016. However, low correlations between years indicated that year × genotype played a substantial role in

ion concentrations. These higher levels of ions in 2016, likely contributed to the increased number of SNPs detected in 2016 compared to 2015. In the V×T population, P, B, and Cu concentrations were within previously reported "normal" limits for *V. vinifera* petioles (*Bates & Wolf, 2008*), had no correlation with observed physiological symptoms, and low to moderate variability among individuals. All other nutrients or ions evaluated were outside of normal ranges or baseline levels have not been established. Concentrations of N, Mg, Na, Fe, and Al were outside (higher or lower) of the normal range for grape and were strongly associated with symptoms in both 2015 and 2016. Deficiencies or surplus of several of these ions can result in chlorosis, marginal leaf burn, or stunting. However, the symptoms observed were not consistent with any single nutrient imbalance or "acidic soil sickness", a term used to describe foliar symptoms related to deficiencies in Ca, Mg, or P from low pH soils (*Wilcox, Gubler & Uyemoto, 2015*). This suggests the symptoms in the V×T population were the result of misaccumulation in more than one ion.

Iron deficiency and aluminum toxicity can result in interveinal chlorosis and necrosis, but not the marginal leaf burn, stunting and chlorosis observed in the V×T population. In our work, strong positive correlations (>95%) were observed among Fe and Al concentrations in both symptomatic and asymptomatic plants across years. A similar positive correlation was detected in maize, but has not been reported in other crops (*Hoffer & Trost, 1923*). Previous studies have shown that, aluminum tolerance variability exists among grape cultivars, with highly sensitive genotypes showing reduced root growth (*Cancado et al., 2009*). Conflicting information exists on the effects of aluminum on accumulation and distribution of nutrients in plants. It has been shown that it can negatively impact plant health by restricting the uptake of nutrients predominantly Ca and Mg in maize (*Mariano & Keltjens, 2005*). However, other studies on maize have shown that Mg and Ca content in the shoots show little variability when exposed to Al in the soil (*Lidon, Azinheira & Barreiro, 2000*; *Olivares et al., 2009*). In our study, high concentrations of Mg were observed despite the high concentrations of Al also being present.

In grape, Mg deficiency symptoms are typically interveinal chlorosis starting at the leaf edge. Mg overaccumulation has not been described in grape, but in other plant species was characterized by stunted growth and foliar yellowing. In our population, marginal, but not interveinal, chlorosis and stunting were observed and positively associated (32–60%) with an increase in foliar Mg content. In excess, Mg can inhibit the absorption of other essential nutrients such as Ca or K affecting root and shoot growth (*Kobayashi, Masaoka & Sato, 2005*; *Tang et al., 2015*; *Venkatesan & Jayaganesh, 2010*). This was similar to our study, where calcium and manganese levels decreased while Mg concentration increased in symptomatic plants. SNPs associated with Mg accumulation were identified on chromosome 3, but none of the genic SNPs were associated with putative transporters and a small 48 bp block of SNPs were not located in a known genic region. A previous study by *Mandl et al. (2006)* determined that Mg deficiency was associated with a region on chromosome 11. In our work, chromosome 11 was associated with Na, but not Mg accumulation. These data combined would suggest that Mg accumulation in the V×T population is not a result of an overexpression of a Mg-specific transporter as was postulated by *Mandl et al. (2006)*. SNPs associated with foliar symptoms were also

predominantly located on chromosome 3, suggesting that Mg content had a role in the visible symptoms. However, many of the remaining symptom-related SNPs did not overlap with those associated with Mg content indicating that this is only one small piece of the equation.

In grape, Na stress symptoms can include internode and leaf stunting, as well as leaf burns (*Sinclair & Hoffman, 2003*). Leaf chlorosis, observed in our study, is not considered a symptom of salt stress in grape, but Na levels were consistently associated with symptoms in years 1 and 2 (*Baneh, Hassani & Shaieste, 2014*). Strong correlations between Na concentrations and those of Mg, Ca, and N were observed in the first year of this study, but were not consistent across years.

In the V×T population, Na accumulation was found to be associated with SNPs located on chromosome 11 consistent with previous work (*Henderson et al., 2018*) in addition to chromosomes 3 and 18. *Henderson et al. (2018)* and *Wu et al. (2020)* found variability in high affinity potassium transporters (HKT) that could improve exclusion of Na in grape leaves, using interspecific hybrids from *V. champinii* and *V. rupestris* and later in *V. vinifera*. The SNP identified on chromosome 11 and found in 2015 and 2016 in the PN40024 genome did not co-localize to regions with known *Vvi* HKT members, and may be a novel modifier of leaf Na exclusion. In our study, individual SNP (genic and non-genic) effects varied widely. Overlap between Mg and Na concentration-associated SNPs and those associated with symptoms (marginal leaf chlorosis, stunting or both) indicate that symptoms were, in part, tied to the accumulation or mispartitioning of both Mg and Na in the vine. The effect of individual SNPs varied suggesting that nutrient-related symptoms in this population may be the result of interactions of various ions, particularly Al, Na, Fe, and Mg.

Grape has a high level of heterozygosity, and separating genotype errors from minor alleles can be challenging (*Hyma et al., 2015*). As more grape genomes are sequenced, it is quite apparent that genomic inversions and deletions are common among cultivars and the grapevine gene annotation is constantly being modified. Some of the candidate SNPs identified here may associate with currently unannotated genes not present in the Thompson Seedless or PN40024 genomes. Additionally, as 'Gewürztraminer' is an aromatic sport of 'Traminer', which itself is the parent of 'Verdejo', this population is genetically similar to an $F_1$ back cross 1 ($F_1BC_1$). The apparent presentation of symptomatic vines in a 3:1 recessive pattern also suggests that both parents may carry associated genes in a heterozygous state that when combined, produce the undesirable trait. Grape is particularly susceptible to inbreeding depression, and these SNPs may be associated with deleterious alleles of regulatory or genic regions not annotated in sequenced grape genomes. While speculative, it is possible that the wide distribution of many SNPs of varying effects across chromosome 3 suggests this chromosome may by carrying deleterious alleles. While multiple SNPs were identified in this study, additional work is needed to confirm their role in nutrient accumulation. When comparing SNP results between the two genomes used in this study, it was clear that chromosome 3 was a major contributor of the phenotypic variation observed in the V×T population. Similarly, individual SNPs identified in both genomes had high variability in effects on symptoms (leaf stunting and/or chlorosis), with few genes having more than one significant SNP. In the Thompson Seedless genome

annotation, most Na associated SNPs were located in large intergenic regions of the genome. Fewer significant SNPs were detected in the PN40024 genome compared to the Thompson Seedless, as was expected due to the increased filtering in the PN40024 genome. However, in Thompson Seedless, multiple SNPs within a single gene were detected, but not for the PN40024 genome suggesting that higher stringencies of filtering could make the dataset more manageable without losing too many regions of interest. The combination of a low read depth threshold and the absence of genetic mapping could result in genotyping errors, which may be a source of error. These data highlight the importance of genome, annotation and filtering, selection when performing these types of studies.

## CONCLUSION

In summary, we evaluated a *Vitis vinifera* segregating population for micro and macro nutrient accumulation across two years. Broad sense heritability was low for most nutrient concentrations and showed no variability in the population for copper concentration. For nutrients with high variability in the population, this low broad sense heritability is indicative of a large environmental component. This was further evident in that specific nutrient concentrations fluctuated with environmental conditions, vine age or the interaction between environment and individual genotype from 2015 to 2016, though trends were consistent across years. Symptom-associated genic SNPs identified were located in putative stress response-related genes. However, many SNPs identified were not associated within known genic regions. Many of the SNPs associated with Mg accumulation were distributed across chromosome 3 for both of the genomes evaluated. While it is clear that a block of SNPs on chromosome 3 is affecting this trait, this bi-parental population had insufficient recombination in the region to identify associated candidate genes. SNPs associated with Na and Mg accumulation as well as foliar symptoms were identified. However, imbalances in neither of these single ions were able to fully explain the observed symptoms, and the relationship with symptoms varied as the plants aged and other nutrient levels changed. These fluid relationships highlight the complexity of micro- and macro nutrient relationships in perennial crops.

## ACKNOWLEDGEMENTS

The authors would like to thank Elisha Partin and Lindsay Wourms for technical assistance in sample collection and in-field phenotyping. We would like to thank Dr. Konstantin Divilov for assistance with testing BLUPs. Mention of trade names or commercial products in this publication is solely for the purpose of providing specific information and does not imply recommendation or endorsement by the U.S. Department of Agriculture. USDA is an equal opportunity provider and employer. The sequencing was carried out at the DNA Technologies and Expression Analysis Cores at the UC Davis Genome Center.

### Funding
Funding for this project was provided by E. & J. Gallo Winery and the USDA ARS. There was no additional external funding received for this study. Dr. Peter Cousins is a grape breeder for E&J Gallo Winery. Dr. Cousins contributed to study design, data collection, and manuscript review.

### Grant Disclosures
The following grant information was disclosed by the authors:
E. & J. Gallo Winery.
USDA ARS.

### Competing Interests
Dr. Peter Cousins is a grape breeder for E&J Gallo Winery.

### Author Contributions
- Rachel P. Naegele conceived and designed the experiments, analyzed the data, prepared figures and/or tables, authored or reviewed drafts of the paper, and approved the final draft.
- Jason P. Londo analyzed the data, prepared figures and/or tables, authored or reviewed drafts of the paper, and approved the final draft.
- Cheng Zou analyzed the data, authored or reviewed drafts of the paper, and approved the final draft.
- Peter Cousins conceived and designed the experiments, performed the experiments, authored or reviewed drafts of the paper, planted and maintained vines, and approved the final draft.

### DNA Deposition
The following information was supplied regarding the deposition of DNA sequences:
Sequences are available at GenBank: PRJNA615210.
Data is also available as a Supplemental File.

### Data Availability
Data is available in the Supplemental File.

### Supplemental Information
Supplemental information for this article can be found online at http://dx.doi.org/10.7717/peerj.10773#supplemental-information.

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
