# Peer review of "Identification of SNPs associated with magnesium and sodium uptake and the effect of their accumulation on micro and macro nutrient levels in Vitis vinifera"

_PeerJ, doi:10.7717/peerj.10773_

## Round 0.1 · original submission · Major Revisions

All 3 reviewers have suggestions for improvement of the manuscript. In addition to reviewers' comments, please, also mention the method used for measuring nutrient levels.

Please submit a revised manuscript addressing the reviewers' concerns plus accompanying letter explaining all changes in the manuscript.

Reviewer 1 ·

Basic reporting

The manuscript “Identification of SNPs associated with magnesium and sodium uptake and the effect of their accumulation on micro and macro nutrient levels in Vitis vinifera” by Naegele, Londo and Cousins (PeerJ id: 40548) reports the identification of a series of SNPs associated with magnesium and sodium levels in grapevine detected by association mapping in a ‘Verdejo’ x ‘Gewürztraminer’ offspring of 249 offsprings.

This work results novel and interesting, as not much information of the genetic basis of micro/macro nutrients accumulation in grapevine is available. The manuscript is nicely written, with clear and professional English language. Introduction is clear and it provides a nice of overview of the issues shown in the manuscript, and the number of references used is adequeate. Figures and tables are relevant, and appropriate for an adequate discussion.

Experimental design

The research shown in the manuscript is original, and it fits with the Aims and Scope of the Journal. Research questions are well defined to fill an identified knowledge gap, and authors carry out a nice investigation to answer it.

Nevertheless, I found authors should explain some of the methodological decisions taken during their investigation:

- My main criticism deals with the mapping approach used, since their experimental approach (the genetic analysis of a biparental cross) seems more adequate to perform a classical QTL mapping study (via linkage map construction and QTL analysis) than the MLM method used in this work. Authors should discuss the reason of this choice, and how this method might have affected the results shown in the manuscript. Ideally, authors should perform a classical QTL mapping analysis to compare if their approach is equivalent to the one shown in the manuscript.

- The second issue I would like to remark is the use of two genomes for reads alignment. Although the advantage is clear (and it is stated in lines 254-256), it is not clear if authors compared and merged the SNPs detected with these genomes (to construct a consensus dataset with common and specific SNPs), or if they conducted MLM association tests with both datasets in an independent way, which in turn will cause some redundancy in the results shown in the manuscript (see Tables 4 and 5). If that is the case, authors should show the number of common SNPs detected using both genomes, which should be high according to the findings of Genova et al., 2014 and Patel et al., 2018.

- Related to the previous comment, did authors discard SNPs mapped the ChrUnk for MLM association tests? (they are not present in the Manhattan plot figure).

- Also related to the previous comment, authors should discuss the different number of SNPs detected in the VxT progeny when using PN40024 or Thompson Seedless genomes for alignment.

- An important point is the low number of markers obtained after GBS. Authors report the detection of “only” 10K/4K SNPs in a Vitis vinifera x Vitis vinifera progeny of 249 offsprings. Recent works report denser datasets. As an example, Tello et al. 2019 report the detection of an average of 17K SNPs in a series of progenies of 60 individuals. Do authors have any reason for such low efficacy? Did authors include Verdejo and Traminer in the GBS approach?

- Some other detected issues include:

o Abstract, line 21. Substitute Vitis for Vitis vinifera L. It applies to the rest of the document.
o The term GWAS should be avoided in the document, as it is used for the genetic analysis of core collections or diversity panels (see line 213, as an example).
o Line 361: genic “deserts”. Do authors mean intergenic regions?
o Lines 378-380. Verdejo Blanco is not the grandson of Traminer. Verdejo Blanco is a direct offspring of Traminer (Castellana Blanca x Traminer). See: http://www.vivc.de/index.php?r=passport%2Fview&id=12949
o Table 1. Authors found higher concentration levels for most nutrients than the expected “normal” range. Although it can be due to transgressive segregation, did authors analyse nutrients concentration in Verdejo and Traminer?
o Tables 4 and 5. The likely role of these genes on the associated trait should be further discussed in the manuscript.

Validity of the findings

GBS are not available and bioinformatic scripts have not been provided (at least, it is not indicated in the manuscript). Authors should solve this issue.

Reviewer 2 ·

Basic reporting

No comments

Experimental design

Sounds very good.

Validity of the findings

No comments

Additional comments

Manuscript from Naegele et al. uses GWAS and identifies SNPs associated with magnesium and sodium uptake and the effect of their accumulation on micro and macronutrient levels in Vitis vinifera. It is a well-written manuscript and presents some useful data for the grape breeding and genetics community so I suggest it should be accepted for publication. However, before its acceptance authors should address some concerns.

I did not find a strong rationale for selecting some micro and macronutrients in the study. I would have loved to see authors talking about it from a biofortification perspective. The authors should focus little more to explain why they selected these 4-5 micro/macronutrients.

These variations in the correlation between the data collected from two different years. For example authors reported “ Foliar symptoms were consistent (≥ 90%) between years and correlated with changes in nutrient concentrations of magnesium (r = 0.65 and r = 0.38 in 2015 and 2016,respectively), aluminum (r = 0.24 and r = 0.49), iron (r = 0.21 and r = 0.49), and sodium (r= 0.32 and r = 0.21)”. I see huge variations in the correlation data between the years. How do authors explain these variations?

Authors have used terms such as Sodium-associated SNPs, I am very critical of such descriptions. Authors should at least mention whether or not these SNPs are associated with higher or lower sodium / or another micro/macronutrient concentrations. It will provide a smooth reading to their audiences.

Reviewer 3 ·

Basic reporting

This manuscript reports a genome-wide phenotype-genotype association study for traits related to plant mineral nutrition in a bi-parental population of cultivated grapevine. Provided that ion measurement protocol is valid (which I cannot assess because I am not a physiologist), this study is sound, producing a large amount of simultaneous results on the genetic variation and determinism of several ions content, with rigorous analyses and interpretation of results. However, it suffers from some weaknesses, which are listed below. Therefore, I recommend major changes to be made before acceptance.

On the whole, the manuscript is written in a clear, unambiguous, professional English language, but it still has to be double-checked to correct some small errors (e.g. lines 224 : 2015 written twice ; 236 : one genes ; 237 : which are a class ; 264 : on chromosome 3, 4, etc).

The context is well described in the Introduction. However, the reporting of previous QTL studies lacks precision:
Lines 88-91, formulation is unclear (“approximately” should be “up to”, important precisions such as “inter-specific cross”, “as rootstock” and “under chlorosing conditions” should be added, and many more small QTLs were found than stated).
Line 94: should be “Henderson et al 2018” and “the author characterized the proteins encoded by 4 alleles of a gene”.
Lines 87-98: all three studies were on inter-specific crosses, which should be stated.

Structure conforms to PeerJ standards, except that “Materials and Methods” section has “Methods” as title, there is no “Conclusions” section, and no separate “Funding statement” section.

Main figures are relevant and high quality, but Figure 2 is not cited in the text and its legend lacks some precisions (remove “foliar” and define the horizontal line). Add “vinifera” to the legends of both figures. Supplemental Figures are not very well explained. Supplemental Figure 1 does not show the same list of ions as mentioned in the text lines 197-198. Supplemental Figure 2 has no legend at all. For marker-trait associations, Manhattan plots should be provided for all traits analysed as Supplemental Figures, for both genomes, in addition to Figure 2.

Raw data are not supplied.

Experimental design

This manuscript reports original primary research within the scope of the journal.

The research question (genome regions associated with variation in grapevine leaf micronutrient content) is well defined, relevant, and meaningful. Correlation between nutrients and deficiency symptoms is extensively discussed, therefore it should also appear in the research question. How this research fills a knowledge gap (on joint genetic variation and determinism of several nutrients in grapevine) should be more clearly stated in the introduction.

Rigorous investigation was performed to a high technical and ethical standard (as far as genotyping and statistical analyses are concerned). The authors chose to detect QTLs with GWAS instead of interval mapping, as usually done in bi-parental populations. This interesting strategy is made possible by the high marker density reached by next generation sequencing methods. It requires no preliminary genetic mapping, and therefore makes it more straightforward to detect correlations with SNPs found with different reference genome sequences. The advantages of this strategy could be emphasized in the manuscript, and its potential limits (uneven marker genome coverage) discussed.
A few minor issues on methods are:
Line 139: Pearson’s correlations on raw or transformed variables? If on raw variables, then rather use Spearman, since raw distributions were not normal.
Line 151: why was read length trimmed at 20 bp? This seems very short compared to usual read length (100-150 bp).
Line 157: a depth threshold of 5 is very small for a heterozygous species such as grapevine; it could lead to inflated genotyping error rates. This should at least be discussed as a potential limit of the study, since no genetic mapping was performed, which would allow to remove markers with genotyping problems.

At many places, methods are not described with sufficient detail:
Lines 119-120: the time and way these nutrients were applied should be described in more details, according to what is stated lines 44-46.
Line 122-123: it is not clear here whether symptoms were measured separately or combined (it is clear only later, at lines 134-135).
Line 124: more details are needed on the number of leaves taken per vine and their development stage, according to what is stated at lines 291-292. In particular, was leaf sampling done exactly the same way in both years?
Line 125: it is not clear here if on the 249 offsprings or on the subsets described in the preceding sentence.
Line 125: air dried: at which temperature and how long?
Line 126: the methods used to quantify each ion are not given here, whereas it is a key point for this study. Please detail, or at least give method names and references. In addition, please explicit whether there were any controls and/or replicates.
Line 133: the statistical method used for testing normality should be given.
Line 134: precise what effect was tested in ANOVA.
Line 137: differences in nutrient concentrations between what?
Line 139: Ward method: which software, version and parameters?
Lines 140-142: not clear at all (“mean square values across years”); please give the page referred to in Fehr 1987, and explicit the formula used for calculating heritability.
Line 148: which enzyme?
Lines 148-149: give a reference for the protocol.
Line 153: replace “Pinot noir” by “PN40024” throughout the text.
Lines 159-161: was association with SNPs tested for all ions measured? It is not clear because distribution and correlation results are given for all 13 ions (Supplemental Table 1), but QTL results are given only for Mg and Na (Supplemental Table 3). Please state here explicitly and complement results and discussion if need be.
Line 156: “using a reference genome”: not clear; do you mean “between reference genomes”?
Line 160: give Tassel version.
Line 166: give Blast2GO version and NCBI database versions.

Validity of the findings

All analysis results underlying conclusions have been provided. Many valuable results are shown. However, additional results of interest could easily been shown and a few points have to be made concerning results presentation.
- Plots and correlations between years are not shown.
- Authors could also easily estimate BLUPs of genotypic values for all offsprings, after fitting a mixed linear model with genotype as random effect and year as fixed effect (phenotype = genotype + year + error). This would allow them to study genetic correlations between traits.
- Raw distributions of traits should be given as supplementary figures and commented.
Lines 158-159: only the total number of SNPs is given. Since no genetic map is produced, the physical distribution of these markers along the genome should be shown, for the reader to assess genome coverage quality.
Line 191-192: it is not clear why heritability is not estimable for these ions? Given the experimental design, heritability should be estimable for all ions quantified.
Lines 197-210: correspondence between text and Tables should be double-checked in this paragragraph:
- Line 199: r up to 0.6146 and 0.4805, not >=
- Lines 199-200: no for Al, OK for Mg
- Line 206: in 2015, not 2016
- Line 207: S P K in 2015, not in 2016
- Line 208: not true in 2016
- Lines 208-210: P/K not in Supplemental Table 2; also comment other ratios
For marker-trait associations, please give in text the total number of significant SNPs for each trait from the start (e.g. line 225: 21 SNPs), and try to more clearly highlight overlaps between years and between genomes (Manhattan plots, as required, would be helpful).
Lines 251-268: not very clear; more clearly separate results on stunting vs combined, and on significant SNPs vs significant SNPs in genes.

Conclusions given are generally well stated, linked to original research question and limited to supporting results. However, a few issues should be addressed and several additional points should be discussed, given the results:
- Discuss differences between years both in ion concentrations, correlations, and SNPs associated (more in 2016): could they partly be due to differences in nutrient supply between years? And/or to genotype x year interactions?
- Discuss low heritability values: could they partly be due to methods (leaf sampling, precision of ion quantification methods)?
- Present results have been obtained in a pure V. vinifera population; discuss what differences this may make with previous works on inter-specific crosses? What is the impact on mineral nutrition of studying offsprings as seedlings/cuttings vs as rootstocks?
- Line 295: the results of Bates and Wolf 2008 are used as reference by the authors; please discuss to what extent they are comparable to your own data (plant management? Sampling procedure? Measurement method?).
- Lines 349-350: results were compared with other GWASs but allele frequencies are not comparable between bi-parental populations and diversity panels
- Line 369: year effect might be due not only to age, but also to environment or genotype x year interactions.
- Line 374: in bi-parental populations, confidence intervals are large due to large extent of linkage disequilibrium, and therefore they harbor many candidate genes.
- Line 381-382: not clear; segregation distortion could easily be tested along the genome (physical positions) for this population.
- Line 384-385: to clarify relationships between symptoms and nutrients, binary regression of symptoms on nutrients could be performed. PCA of nutrient contents (in each year and/or with inter-year genotypic BLUPs) may also help clarifying the relationships between these traits.

Additional comments

Miscellanous:
In the abstract: add main methods (GBS and GWAS) and the range of QTL effects (% variance explained by SNPs)
Table legends: often incomplete to be self-sufficient (e.g. for Tables 2 and 3, “in a Vitis vinifera F1 population” should be added)
Replace “Thompson” by “Thompson seedless” throughout
Lines 44-49 should be moved to line 75, and V. vinifera should be presented as the cultivated species at line 75
Line 145: replace “progeny” with “offspring”
Line 247: “similar replications” between years?
Line 254: replace potential by total
Line 265: “significant SNPs” in genes?
Line 307: >95%?
Line 347: give reference just after “previous work”
Table 1: replace “Normal range” by “Reference range”
Table 3: why is Ca in this Table and not only in Supplemental Table, since it is not correlated with symptoms in both years? Replace “Multivariate analysis of correlation” by “Correlations”, since only 2-way correlations were tested.
Table 3 and Supplementary Table 1: highlight strong correlations (r > 0.3) for ease of reading.
Table 5 is cited before Table 4 in text.
Table 5 line 7: add “and NCBI”
Supplemental Table 3: why are there negative R2 values?

---

## Round 0.2 · Minor Revisions

The revised manuscript has improved, but 2 reviewers still request further minor revisions.

Reviewer 1 ·

Basic reporting

The reviewed manuscript “Identification of SNPs associated with magnesium and sodium uptake and the effect of their accumulation on micro and macro nutrient levels in Vitis vinifera” by Naegele et al. (ID: peerj-40548-2) reports the identification of some SNPs associated with magnesium and sodium levels in grapevine detected by association mapping in a ‘Verdejo’ x ‘Gewürztraminer’ offspring of 249 individuals. I appreciate the answer given by the authors to all my suggestions, and the effort done to improve the quality of their manuscript. Therefore, I only have some minor suggestions.

L. 110. Material and Methods.
When transplanted, were the seedlings grafted or were they grafted on their own roots?
L. 152. The R package (and version) used for BLUP values calculation should be specified.
L. 178: Replace “principle components” by “principal components”.
L. 218. Authors found that: “Na, P, Cu, Mn, N, S, and Ca concentrations in 2015 or 2016 had no heritability”. Authors should discuss the power and precision of the reported QTLs in LGs 11, 12 and 18 for Na concentration considering that they suggest it as a non-heritable trait.
L. 225. Figure 2 does not show the correlation between symptoms and nutrients variation. Please, correct.
L. 229. How is this 22% value estimated?
L. 232. In my opinion, a correlation r=0.3 can not be considered as “strong”. In fact, in line 236 authors refer to correlations r= 0.2-0.4 as, low-moderate, which is more realistic.
L. 237. Why is copper shown in capitals?
L. 243. Replace “Significant positive associations” by “Significant associations”.
L. 245. Rewrite “Mg-related SNP effects were approximately 6%” by “SNPs associated with Mg levels explained ca. 6% of trait variation”, or something similar.
L. 256. “SNPs […] had both negative and positive effects” Per se, SNPs do not have positive nor negative effect on traits. Alleles do. Please, correct.
L. 258. “Two of the SNPs detected (S11_16152632 and S18_25745143) were detected in 2015 and 2016”. Please, rewrite.
L. 260. “Only a single SNP identified in 2016 (S11_11635675) was also 261 associated with a gene”. Do authors mean that only one SNP identified in 2016 was found in a gene region?
L. 261. “11 positive effect SNPs”. Please, correct as suggested for L.256 and check the whole document.
L. 290. Figure 2 is a PCA plot, not a Manhattan plot. Please correct and check the whole document for similar errors.
Table 5. “Pinot Noir 12x” Authors must avoid naming the reference genome as “Pinot Noir”, as it is not. The reference genome they refer to is the Vitis vinifera cv. PN40024, a nearly homozygous line derived from Pinot Noir. They should check the whole document.
Table 6. Replace “Thompson Sdls” by “Thompson Seedless”. Authors should check the whole document to harmonize the name of the Thompson Seedless cultivars, sometimes referred as “Thompson Seedless” and sometimes as “Thompson seedless”.
Figure 1. I think the symptomatic (stunted) stem internodes they refer to are shown in Figure 1B, not 1A. Please, correct. Besides, Figures 1C and 1D are not too clear.

Experimental design

no comment

Validity of the findings

Authors should discuss the power and precision of the reported QTLs in LGs 11, 12 and 18 for Na concentration considering that they suggest it as a non-heritable trait.

Reviewer 3 ·

Basic reporting

Most points raised by the reviewers have been answered satisfactorily. However, revision is still needed for the following points.

I cannot see any indication that raw data are supplied in the beginning of the submission and I cannot see any " important announcement " section in the revised manuscript. If GBS data have been made available, this should be written in the text, with the corresponding links (I did not manage to find them in NCBI). Moreover, plant identifiers in the vcf file do not match plant identifiers in the phenotype file, thereby preventing duplicating the analysis. Please provide the correspondence between both identifier lists.

Some Table and Figure numbers have not been corrected in the revised text, please double-check numbering (e.g. Line 251 : replace " Suppl Table 3 " by " Suppl Table 2).

Figure 3 : is this Figure for symptoms (both stunting and chlorosis, as stated in Figure legend) or for stunting only as stated in the text ?

" We have however performed an additional PCA analysis of ions to demonstrate the weighting of Mg, Fe, and Al specifically along the vector where symptomatic plants occur. (Figure 2). " : this PCA is interesting (provided they are based on BLUPs of genotypic values), but symptomatic plants are not projected on it.

In supplemental Table 3, only the diagonal is of interest.

Line 275 : on chromosomes 11, 12, 13, and 18 (see Suppl Table 5). Please double-check consistency between Tables and text throughout.

Experimental design

The reasons why the authors chose to apply GWAS rather than QTL detection are not given in the discussion, unlike stated in the answer letter. Moreover, line 364 : Zou et al 2020 did not perform QTL detection, but only genetic mapping.

Several comments concerning BLUPs:
- Lines 163-165 : the authors have estimated BLUPs, but it is not clear at all what linear model they used, since it is not stated that they have included genotype effect as a random effect. The classical full model to use is : phenotype = genotype (random) + year (fixed) + error (random), where genotype is usally included but year is not always included, depending on the trait. Please clarify.
- All BLUP values do not need to be given as a Supplementary Table.
- " high correlation between average ion concentration and computed BLUPS indicate this method does not add much to the current analysis. " : BLUPs of genotypic values are more precise estimates of genetic values only when year effect is significant.
- " BLUPs have been calculated and are included as a supplemental file. However, these results did not differ from our previous measures of correlations between ions and symptoms. " : correlations between BLUPs of the different traits (genetic correlations) are not shown.
- Line 166 : " the mean BLUPs " : not clear.

" Line 134: precise what effect was tested in ANOVA." : still not precised line 152.

Validity of the findings

Lines 375-376: Replace " However, correlations between years indicated that year x genotype played a substantial role in ion concentrations " by " However, low correlations between years indicated that year x genotype played a substantial role in ion concentrations "

" A depth threshold of 5 is very small for a heterozygous species such as grapevine; it could lead to inflated genotyping error rates. This should at least be discussed as a potential limit of the study, since no genetic mapping was performed, which would allow to remove markers with genotyping problems. " : All I can find in the discussion to answer this point is " Grape has a high level of heterozygosity, and separating genotype errors from minor alleles can be challenging (Hyma et al., 2015). " I insist that it should be explicitely stated that the combination of the low depth threshold and the absence of genetic mapping may result in genotyping errors, which is a limit of this study.

" Text has been updated to explicitly state that all ions were tested, but no QTL detected except for those described. " : I cannot see where it is stated that " no QTL were detected except for those described "

Lines 235-236 : still not clear : had null heritability ?

Lines 243-244 : still not exact : replace " In 2015 and 2016 concentrations of Mg (r = 0.6146) and Al (r = 0.4805) " by " In 2015 and 2016 concentrations of Mg (r <= 0.6146) and Al (r <= 0.4805) "

Lines 243-245 : replace " In 2015 and 2016 concentrations of Mg (r = 0.6146) and Al (r = 0.4805), respectively had the highest correlation with observed vineyard symptoms (Table 3) " by " Mg concentration in 2015 and Al concentration in 2016 had the highest correlation with observed vineyard symptoms (r = 0.6146 and 0.4805, respectively) (Table 4) "

Lines 255-257 : not clear which test was applied ; moreover, some ratios were consistently associated with symptoms in both years

For the points listed below, I can see no change in the v1 pdf version. Please check which version of Tables you included into the pdf.
- " Table 1: replace "Normal range" by "Reference range" - text has been adjusted "
- " Table 3: why is Ca in this Table and not only in Supplemental Table, since it is not correlated with symptoms in both years? Replace "Multivariate analysis of correlation" by "Correlations", since only 2-way correlations were tested. - Ca was deleted from Table 3 and title has been changed. ",
- "Table 3: highlight strong correlations (r > 0.3) for ease of reading. - Tables have been bolded."
- "Table 5 line 7: add "and NCBI" - text has been adjusted.

There were indeed negative R2 values in former Suppl Table 3 (for example, S3_5986778 had R2 value -0,054992312) but only for PN40024, but the equivalent for PN40024 of revised Suppl Table 5 is no more available in the revised version. In revised Suppl Table 5, what does " SNP effect " stand for ? Line 276 it is stated that " effects ranged from 8 to 13% of the observed variation ", but there are both positive and negative values in the " SNP effect " column.

---

## Round 0.3 · Minor Revisions

Please, correct typo at line 428:
"Gewürztraminer"

---

## Round 0.4 · Minor Revisions

There were some comments of the section editor of PeerJ, which should be addressed:

"A number of things need to be addressed: Figure 3 legend says significance threshold is 1X10-7 but the line is drawn at 1X10-6.

For readers used to working with inbred strains the study design is confusing because F1s from inbreds would all be genetically identical. I assume this study works because the parents are not inbred; can this info and the reasoning behind the mapping population design be added somewhere?

The SNP density is very low for a GWAS but probably OK here because of the study design. Compute average extent of LD to confirm +++ The fastq files in the raw data set are extremely incomplete (only ~250 reads per file)

For the fastq files to be useful the barcoding/indexing key needs to be provided.

Ideally fastq files would be deposited at NCBI Short Read Archive.

Supplemental Figures need to be organized with page breaks between figures so that the legends are actually with the figures, for example the legend for Supp Fig 3 is directly beneath Supp Fig 2.

Supplemental Figure 2: each row is an F1? Colors represent? labels are too small

Supplemental Figure 3: can't read the labels, + need sign threshold.

Supplemental Figure 4: text too small."

---

## Round 0.5 · accepted · Accept

In the final version of the article, correct:

line 432: Gewuerztraminer